# Male Partners’ Roles in Infant Feeding Practices: Perspectives of Black Mothers Living with HIV in Three Countries

**DOI:** 10.3390/healthcare10112254

**Published:** 2022-11-10

**Authors:** Josephine Etowa, Hilary Nare, Vuyiseka Dubula-Majola, Olaide Edet, Mildred John, Chioma Rose Nkwocha, Colleen Stephens, Nokwanele Mbewu, Jean Hannan, Egbe Etowa

**Affiliations:** 1School of Nursing, Faculty of Health Sciences, University of Ottawa, Ottawa, ON K1H 8M5, Canada; 2Africa Centre for HIV and AIDS Management, Faculty of Economic and Management Sciences, Stellenbosch University, Private Bag XI, Matieland 7602, South Africa; 3Department of Nursing Science, University of Calabar, Calabar 540242, Nigeria; 4Department of Nursing Science, University of Port Harcourt, Choba 500102, Nigeria; 5College of Nursing, Florida International University (FIU), Miami, FL 33199, USA; 6Philani Maternal, Child Health and Nutrition Trust, Cape Town 7791, South Africa; 7Nicole Wertheim College of Nursing and Health Sciences, Florida International University, Miami, FL 33199, USA; 8Daphne Cockwell School of Nursing, Faculty of Community Services, Toronto Metropolitan University, Torontom, ON M5B 2K3, Canada; 9Ontario HIV Treatment Network, Toronto, ON M4T 1X3, Canada

**Keywords:** black mothers living with HIV, infant feeding practices and guidelines, male partners’ role, psychosocial distress, social support

## Abstract

Currently, mothers living with HIV (LWH) are challenged with different infant feeding guidelines depending on the country they are living in. This may contribute to confusion, stress, and mental health issues related to decision-making about infant feeding as a mother LWH. Yet, their male partners as their closest social capital have important roles to play in reducing or aggravating this psychosocial distress. Hence, we describe the role of male partners in supporting mothers who are living with HIV in the context of infant feeding. It is based on the results of a recent study of the socio-cultural context of infant feeding among Black mothers LWH in three countries; Canada, the USA, and Nigeria. The study was a tri-national, mixed-methods, community-based participatory research (CBPR) project, informed by postcolonialism and intersectionality theories. This paper is based on the qualitative component of the study. It was a focused ethnography (FE) involving 61 in-depth individual interviews (IDIs) with Black- mothers LWH. Thematic analysis guided the interpretation of these data, and trustworthiness was established through member-checking. Black mothers LWH acknowledged the various support roles that their male partners play in easing the practical and emotional burdens of infant feeding in the context of HIV. Male partners’ roles were captured under three sub-themes: (1) Practical help, (2) Protection of the family, and (3) Emotional support and sounding board. These findings have explicated the evolving ways in which male partners support ACB mothers LWH to promote positive infant feeding outcomes, as well as enhance the emotional and physical well-being of both mother and infant. Our study has explicated the evolving ways in which male partners support Black mothers LWH to promote positive infant feeding outcomes, as well as enhance the emotional and physical well-being of both mother and infant.

## 1. Introduction 

Available statistics show that in sub-Saharan Africa, which bears 67% of the global burden of HIV, 15.9 million women (vs. 9.8 million men) are living with HIV (LWH), and adolescent girls and young women (aged 15 to 24) accounted for 25% of new HIV infections in 2019, despite making up only 10% of the region’s population [1]. In the USA, the Centers for Disease Control and Prevention (CDC) reported that, while Black women composed only 13% of the U.S. female population, they accounted for 58% of women diagnosed with HIV in 2018 [2,3], highlighting the disproportionality of Black women among women LWH. Canadian data from provincial and territorial surveillance programs show that, while Black men accounted for 17.9% of new HIV cases among men in 2017 (vs. 41.7% for Caucasian men), Black women were the highest group among women, accounting for 46.3% (vs. 14.1% for Caucasian women) [4]. Furthermore, within the group of perinatally HIV-exposed infants, Africa was the maternal region of birth of 38.6% of infants, and 50% of the infants exposed were Black (vs. 23.3% for Caucasian infants and 18.1% for indigenous infants), further emphasizing the health inequities which the offspring of Black women face starting in the womb.

Mothers LWH are advised by World Health Organization (WHO) Guidelines to modify their infant feeding decisions to reduce the risk of mother-to-child transmission (MTCT) of HIV. For mothers LWH on lifelong antiretroviral therapy (ART) in resource-constrained countries, the 2010 WHO Guidelines recommended exclusive breastfeeding (EBF) for the first six months, followed by six months of breastfeeding and complementary feeding [5]. The updated 2016 Guidelines encouraged EBF for a minimum of 12 months, followed by breastfeeding and complementary feeding for up to two years, or longer if possible [6]. 

Nonetheless, in many sub-Saharan African countries, where mixed feeding is the cultural norm, EBF is often seen as an indicator of HIV infection, prompting many mothers LWH to abandon it in favor of mixed feeding to avoid stigma and to comply with encouragement and pressures from relatives and neighbors [7,8,9]. In Tanzania, for instance, the EBF rates among mothers LWH sharply declined from 85% to <30% between the second and fourth months post-partum [10]. In Zambia, where HIV affects approximately a fifth of pregnant women, 26% of mothers LWH never initiated breastfeeding and 55% were not breastfeeding by 6 months post-partum [8]. Similar trends were observed in Burkina Faso and Cameroon [11], Nigeria [12], as well as in a South African study, where 80% of mothers LWH who had chosen EBF switched to mixed feeding within the first month [13].

In contrast, national infant feeding guidelines in the U.S. and Canada prohibit mothers LWH from breastfeeding and require the exclusive use of formula feeding (EFF) in a bid to curb vertical transmission [14,15,16]. However, EFF overburdens diasporic ACB mothers living with HIV (ACB-Mothers LWH), as they get caught up between EFF national guidelines, a local culture of “breast is best”, and the differing cultural expectations of their extended families and countries of origin. ACB mothers LWH particularly fear that EFF would inadvertently disclose their HIV status [17].

The higher challenges which Black mothers LWH face in the context of infant feeding can be mitigated through a collaborative effort with male partners, and the literature documents some examples in support of this statement. For instance, male partners of Black mothers LWH played an influential role in the selection [18] and maintenance [19] of infant feeding choices, visited clinics to obtain infant feeding formula [20], provided emotional support to ease the mothers’ distress and pain during breastfeeding [21,22,23], fended off pressures to mix-feed from extended family unaware of HIV infant feeding guidelines [24], and even relocated the mothers in the face of significant societal pressures to shield them from the stigma associated with unconventional infant feeding choices [25]. In addition, research in South Africa showed the increasingly visible involvement of male partners in infant feeding decisions and their advocacy for the use of free formula milk from public health facilities over breastfeeding to minimize the risk of MTCT [18]. Therefore, the male partners of Black mothers LWH have already proven to be instrumental in the successful adherence to infant feeding guidelines and the prevention of MTCT of HIV.

In contrast, other studies in Uganda [26] and Namibia [27] revealed that fathers and male partners were rarely involved in newborn care due to gender-based cultural norms and roles, as well as working outside the home [28]. In rural Kenya, even when fathers did not attend infant feeding counseling, mothers LWH often lacked decision-making autonomy in terms of infant feeding, as the infants were generally perceived as “belonging” to the father, which presented a major challenge in the control and prevention of MTCT [29]; and although mothers were responsible for feeding, when male partners were involved, the males’ decisions were final as is customary in some African cultures. In the U.S., while African American male partners may support the mothers in the absence of extended family, male support was more likely to be in the form of spending time playing with the infants than feeding or cleaning them [21,30], demonstrating how African culture and gender-based roles are carried over to diasporic ACB communities.

While previous research findings highlighted the various ways in which male partners support breastfeeding Black mothers LWH, our research extends this knowledge to highlight the role of male partners/fathers in supporting mothers living with HIV in the care and decision-making surrounding infant feeding irrespective of the methods; breastfeeding or formula feeding. Exclusive breastfeeding (EBF) programs are twice as likely to be successful when fathers and male partners are involved [31]. This points to the need for research into male partner involvement in the HIV care and prevention necessary to reduce vertical transmission (mother-to-child transmission/MTCT) of HIV. This paper contributes to existing knowledge with regard to the involvement in infant feeding and care of diasporic and indigenous African male partners and the extent to which they provide support to Black mothers LWH.

## 2. Methods

This study is based on the qualitative component of a larger mixed-methods study. This broader study was guided by postcolonialism and intersectionality frameworks and was implemented as a community-based participatory research (CBPR). Postcolonial theory is an arm of critical social theory that focuses on the lingering effects of colonialism and other forces of oppression by colonized peoples [32]. This facilitated a close examination of African, Caribbean, and Black (ACB) mothers’ experiences of infant feeding while living with HIV. ACB mothers were simply referred to as Black mothers in the study. Intersectionality theory accounted for the understanding of “how multiple social identities such as race, gender, sexual orientation, SES, and disability intersect at the micro level of individual experience to reflect interlocking systems of privilege and oppression” [33].

It provided a way of understanding and analyzing how structural factors interact to produce specific health outcomes for individuals such as Black mothers. CBPR is an innovative research paradigm that combines knowledge and action to improve community health and reduce health disparities [34], and is well aligned with intersectionality theory to support a critical analysis of how ethnicity and gender intersect during infant feeding decision-making by Black mothers LWH. The qualitative component of this study which is the basis for this paper was a focused ethnography (FE) involving in-depth individual interviews (IDIs) with sixty-one Black mothers LWH. Within the ethnographic approach, which traditionally describes entire cultures and communities [35], our research aligns with Mayan’s FE, where a specific question is investigated among a small group of people within a strictly defined context or organization [34].

### 2.1. Setting

This three-country study was conducted in Ottawa, Canada, Port Harcourt, Nigeria, and Miami, USA. All mothers in Port Harcourt were of Nigerian origin and residence, while mothers in Canada and the USA either had a long ancestral history in North America or had originated from various Caribbean and sub-Saharan African countries. This heterogeneity of our study sample ensured the representation of the different ethnocultural perspectives across the global ACB communities.

### 2.2. Ethical Consideration

This manuscript is based on our original three-country study funded by the Canadian Institute of Health Research (CIHR), Funding Reference # (FRN): 144831. Ethical clearance was sought and obtained from four affiliated universities’ research ethics boards (REB). The study was approved by the Health Sciences and Science Research Ethics Board at the University of Ottawa (certificate #H08-16-27), (Canada), the primary host institution of the study. It also secured REB approval of the study protocol from Carleton University Research Ethics Board-A (CUREB-A, certificate #106300) (Canada), the Social and Behavioral Institutional Review Board at Florida International University (certificate #105160) (USA), and the Research Ethics Committee at the University of Port Harcourt (certificate #UPH/CEREMAD/REC/04) (Nigeria). Additionally, permission was obtained from each of the community partner sites where participants were recruited. Voluntary written informed consent was obtained from all participants. As part of the consenting process, participants were assured that they do not have to answer every demographic or interview question, they could choose whether to be recorded on audio, and they could withdraw from the study at any time. They were especially informed that not consenting would not affect their access to any care or services they were receiving. An interpreter was used during both the consenting and interviewing processes as necessary. The principles of confidentiality and anonymity were always observed, including in the storage of research materials. Prior to any engagement, participants were offered information about the applicable community or HIV/AIDS support services that may be useful if they encountered any instances of extreme distress during the study. All interviews were conducted in a private room in the relevant healthcare institution or support agency, or in a safe, private setting of the participant’s choice.

### 2.3. Research Team

The team was composed of Canadian, Nigerian, and US researchers and knowledge users who have been substantively involved in Black communities. The Nominated Principal Investigator, J. Etowa is a founding member and past president of the Health Association of African Canadians and is an active member of the African Caribbean Council on HIV in Ontario (ACCHO) with an extensive background in HIV and CBR work and good knowledge of the ACB community in Canada and abroad. She was responsible for the overall management of the project. Other team members and site leads have experience and infrastructure in their universities to mentor trainees and offer research training for medical and health sciences students.

### 2.4. Data Collection

Keeping with the tenets of focused ethnography which guided the qualitative component of this study, we interviewed 61 mothers living with HIV during the period between 2016 and 2018 from all three countries of study, Canada, Nigeria, and the USA. Sampling method was purposive [36] and supported by complimentary recruitment methods such as the snowballing technique. The snowballing technique occurs when participants serve as recruiters of potential participants for further participation [31]. Participants were recruited at various locations, including community resource centers, public health facilities, AIDS service organizations, immigrant support agencies and organizations, as well as through physician offices. Recruitment was supported by intermediaries, in many cases with our collaborators and their organizations. 

Guided by the colonial theory and intersectionality, we used a critical perspective to draw the interview questions in a way that responses provided by the mothers and the analysis from those responses provide sufficient insights into how structural barriers and the complex interplay of factors influence their experiences and the supports they receive from their male partners. Additionally, guided by the CBPR process, the interview questions were designed to engage the mothers to strengthen their critical health literacy through the knowledge exchange that takes place in the interview process. 

The interview team included research assistants, graduate students, and research coordinators. They were trained between December 2016 and May 2017 in the conduct of ethical and sensitive research with immigrant populations including the conduct of IDIs, the use of NVIVO in qualitative data analysis, etc. All interviews were audio recorded and transcribed verbatim. The interviews were conducted simultaneously, and the transcripts were first analyzed at the country level and the across-country level of analysis was completed for this paper. Data collection continued until we reached the point of saturation [37] in each study site. To ensure that the study’s main points of interest were satisfied during the interviews, we used semi-structured interviews, with questions organized around a topic guide [38]. To capture the mothers’ perceptions of the roles their male partners played in infant feeding, we asked “How did you decide on an infant feeding method?” and followed up with “Did you consult anyone about the choice?” Probes were used to clarify issues raised by the mothers. 

### 2.5. Data Analysis

The data obtained were identified, classified, and subjected to abstract generalizations and pattern explanation in an undulating and cyclical, rather than linear, pattern. For ethnographic data analysis, we interrogated the data following clearly defined steps as per the principles expounded by Roper and Shapira [39], whose method was chosen by us due to its systematic approach to analysis, as well as its high compatibility with both ethnographic data and qualitative data analysis software such as Nvivo. Analytical steps included: (i) coding for descriptive labels, (ii) sorting for patterns, (iii) the identification of outliers or negative cases, (iv) generalizing with constructs and theories, and (v) memoing, including reflective remarks. Through reflexive team meetings, we achieved confirmability. Additionally, to ensure dependability, we established a clear audit trail and accurate documentation of the research processes and procedures, including the analytical process, field notes, digital recordings, and hard copies of interview transcripts. Finally, to ensure transferability and eliminate biases, the transcripts were carefully verified and checked with care paid to issues of misrepresentation of the data that could occur in the data analysis process.

### 2.6. Demographic Characteristics of Participants

A total of 61 women participated in the study: 30 in Port Harcourt, 20 in Miami, and 11 in Ottawa, with the latter group being either English- or French-speaking. Participants were primiparous or multiparous HIV-positive women of African descent who had given birth at least once in the previous five years after their HIV diagnosis. Purposive sampling was used, though the final sample size was dependent on achieving saturation [40]. We aimed to achieve maximum variation [37,41] with respect to social class, as well as the length of time since the migration to Canada or the USA in the case of diasporic participants. The ethnocultural profile of our sample varied by site to reflect the local heterogeneity of Black women in each country and to generate regionally specific knowledge. 

## 3. Result

The data collected for the most part supports the idea that the positive involvement of male partners in infant feeding is critical for the creation and maintenance of optimal conditions both for the healthy nourishment of infants and for the mental and emotional well-being of Black mothers LWH. The findings showed that Black men were responsible, prioritized family interests, acted as providers and protectors, and were increasingly involved in infant feeding in support of Black mothers LWH. The roles of male partners, as described by Black mothers LWH, the mothers in the study, were captured under three sub-themes: (1) Practical help, (2) Protection of the family, and (3) Emotional support and sounding board. Despite the positivity created by the Black male partners with their involvement in infant feeding, the issue has not been immune to challenges. The data collected also revealed challenges posed by fathers in infant feeding. These were captured under the themes, (1) Fathers’ opposition to mothers’ infant feeding choices, (2) Lack of trust in the male partner, and (3) Feeling the absence of the male partner. 

### 3.1. Practical Help with Childcare and House Chores

Across all three study sites, the mothers valued the practical help that their male partners provided, and often spoke about the partners helping with household chores and participating in childcare, whether in the form of playing with and taking care of infants or caring for older children: 


*Sometimes my husband will just say “Don’t worry, let me go and carry the child.”*
(Mum, Port Harcourt, Nigeria)


*When [my husband] is available, he’d play with [the children], he would take them to the park. He’s like a machine. He has to take them to the park. He has a routine. In his schedule for the day, he has to make time to play with them, like they like [my husband] more. They’re like daddy’s girls.*
(Mum, Ottawa, Canada)

Some participants spoke of partners’ support in the form of assisting with chores around the house:


*[My husband] will always come and support me. He will always come and offer assistance towards, like the chores at home, before we started getting the things [domestic equipment] that will ease me out*
(Mum, Port Harcourt, Nigeria)

### 3.2. Protection of Family

Mothers LWH often spoke of having to navigate intense pressure from family and friends who try to influence or undercut the mothers’ infant feeding decisions, despite not necessarily being aware of the mother’s HIV status or the WHO infant feeding guidelines. Some participants described taking their infant feeding decision jointly with their male partners, and some highlighted how receiving decision support from their partners helped them resist family pressure.


*So I now took the decision with my husband, with the help of my husband, and he was so happy… so he was the one that really encourages me, yes, he’s the one that encourages.*
(Mum, Port Harcourt, Nigeria)

A diasporic mother in Ottawa described how her husband’s support was crucial, as he resorted to making up a story to justify their infant feeding choice and fend off pressures from his family.


*She [mother in law] was like, “Put the child on [the breast]!” and I was like, “No!...The milk dried a long time ago, so I cannot give it to the child.” She called [my husband] … and [he] told her the story about how I had an infection after birth, and how I had to use antibiotics. And she tried to give me things to bring the milk back…And later on, she gave up because there was no way I was going to breastfeed because I was going to protect my children, and nothing would stand in between me and protecting my children, no matter what the family does.*
(Mum, Ottawa, Canada)

An Indigenous mother in Port Harcourt described how forming a united front with her husband helped her to resist unsuitable feeding advice given by her mother-in-law: 


*My mother-in-law came and I was like hiding. She was telling me, “Give food, give water, give this.” My husband called me privately and said, “This child is our child. This woman is an old woman; their time has gone. Let’s do what they said we should do; let’s do exclusive baby friendly [formula feeding] for the child. If this child is negative, it’s our own glory tomorrow, but if the child is positive, the blame self will even come from the woman that even advised you to…”*
(Mum, Port Harcourt, Nigeria)

Some diasporic participants expressed how physical distance from their extended families helped reduce family pressures. An Ottawa-based mother commented:


*No one around me, and we didn’t have anyone [around us]. Just me and my husband and my kids. No pressure [from] any family. I knew that I wouldn’t have any crazy pressure. And that was good.*
(Mum, Ottawa, Canada)

### 3.3. Emotional Support and Sounding Board

During pregnancy, as mothers LWH had to come to terms with how their HIV status might affect their health and that of their babies, participants often spoke of the emotional and practical support offered by their male partners, and how it helped them cope with emotional and other challenges.

A participant described how her husband advised her and the health workers about her situation:


*What contributes is that my husband is behind, rightly behind me, supports me, advise[s] me and health workers. So, I have so many people beside me that always, I don’t tell my things to everybody, either health workers or my husband, so their advice supports me a lot.*
(Mum, Port Harcourt, Nigeria)

Several participants described how crucial their partners’ encouragement and emotional support had been to their mental health and to helping them recover from very low points in particular:


*It was when I was pregnant, I was disturbed, but my husband, he encouraged me.*
(Mum, Port Harcourt, Nigeria)


*And for [my husband], I’m thankful because he got me from one point to another point with me having HIV....I don’t know if I would have lived longer because sometimes when you think about it, it’s not the disease that kills you, it’s the way you take [handle] the disease. Like if you’re not positive, at some point some emotional things will pull you down. Maybe if I had not found [my husband], maybe I wouldn’t be here to tell my story. I don’t think I would be here.*
(Mum, Ottawa, Canada)

While most participants had several sources of support, many identified their partners’ support as the most important:


*I have special people that I can boldly go to them and say, “This is my challenge, what do you advise me?” Then, I will take the advice and act on it. Then, my closest person is my husband. I don’t hide anything from him. I let him know everything…you will even discover that the more you keep on telling him, he will even be the one that will even make you feel more better, so he is my closest person that I relate with.*
(Mum, Port Harcourt, Nigeria)

Some participants felt that their support network was very limited, and that they were depended entirely on their partner: 


*For now, no support. The only support I’m having is me and my husband; that’s all.*
(Mum, Port Harcourt, Nigeria)


*It’s only my husband. Ahhh, my husband is trying. He was really of help for me. If he is around, he will really help. He helped me a lot, so except he travelled, except he’s not in town.*
(Mum, Port Harcourt, Nigeria)

### 3.4. Challenges Arising from the Lack of the Male Partners’ Support

Despite the immense role played by fathers in supporting mothers LWH on infant feeding, the process has not been immune to challenges, as fathers are not always supportive as they might hold different perspectives to those of their partners. This has sometimes left mothers feeling stressed and burdened as they have to take care of the baby and deal with diverse socio-cultural problems emanating from their husbands and the society around them.

Some mothers felt the absence of their baby’s father. For some mothers, their baby’s father was not the first point of call when it came to social support. Mothers had to resort to family members for support while others had to be strong and do everything on their own. 


*“Yeah, I go to my family. Sometimes I call them on the phone and talk to them and, you know. They do not even like my daughter now. My daughter now do not even like my own husband cause she do not like the way how he be doing to me kinda way. So she do not wanna be around him that much”*
—mum, Miami, USA


*“Let us see... Was he supportive? [name of child 2 withheld]’s dad had a drug problem. Basically, I felt like I did everything by myself because he was more chasing his addiction [than] being a father to his child so…that is why I really did not mention him. However, we do have an understanding.”*
—mum, Miami, USA

### 3.5. Lack of Trust in the Male Partner

A mother lost trust in her baby’s father due their seropositive status and for being the route of her HIV infection. This potentially left a void in the role of the father in infant feeding decisions.


*“I got it from my kid’s father because he wasn’t being honest with me.”*



*“The thing was with him, he told me that if he woulda told me, I would have left him. Which is right, I would have left him. But, he didn’t give me no choice. He took my choices *voice cracking*”*
—mum, Miami, USA

### 3.6. Fathers’ Opposition to Mothers’ Infant Feeding Choices

Some fathers opposed the mothers’ choice in favor of cultural beliefs and against the national guideline on infant feeding. Mothers showed that situation was compounded by relatives’ opinions, which also favored cultural norms on infant feeding.


*The father, initial stage was not. He wasn’t supportive, he was telling me he wouldn’t want me to faint or fall down on the way because I’m doing exclusive, that I should stop it. And even some, some of the family relations that came, you know, when a woman gives birth, people come for visitation, they said ‘what kind of new, new generation pattern, that in the olden says it wasn’t exclusive’. Even one of the ladies wanted to give water, I said ‘no, no, no, no’, I nearly fought with her”…*
(Mum, Port Harcourt, Nigeria)

## 4. Discussion

HIV has not only altered infant feeding discourse, but also gender-based roles, especially in Black cultures, where tasks and decisions related to infant care are often viewed as a ‘women’s job’. For instance, studies in Malawi [42] and Tanzania [26] revealed the difficulties associated with encouraging men to partake in infant feeding tasks, as some men viewed such tasks as a woman’s exclusive responsibility, viewed themselves as family protectors and providers, and even regarded themselves as superior to their partners [8,42,43]—all of which are attitudes that undermine the significantly valuable supportive role that male partners of mothers LWH could potentially play to ease the burden of HIV on both mother and child.

In our study, we found that, in the challenging context of infant care while living with HIV, many Black mothers LWH had positive perceptions of the support they received from their male partners. We have categorized and presented their description of this support as: (1) Practical help, (2) Protection of the family, and (3) Emotional support and sounding board.

With respect to practical help, Black mothers LWH noted that their male partners were directly involved in childcare, including feeding, bathing, and playing with the infant, as well as looking after older children. In addition, they acknowledged that the level of involvement depended on the male partner’s working hours and availability vs. absence from the home. This finding conforms with a quantitative analysis from the same broader study showing that majority of the male partners (Ottawa [80.0%], Miami [77.9%], and Port Harcourt [68.0%]) were supportive of the mothers adhering to the infant feeding guidelines. The study concluded that the male partner’s supportive opinion of the national infant feeding guideline was a significant factor in the mother’s adherence [14]. Moreover, there have been other findings where African American fathers from middle- and lower-income families showed high levels of involvement in the caregiving of young children [30,44]. On the other hand, a study found that men’s absence from the home limited their infant feeding support [45]. 

In contrast, the mothers LWH in our study emphasized frequent pressures by relatives (e.g., mother-in-laws) to abandon their infant feeding choices and to conform to culturally normative ones, often in contradiction to WHO infant feeding guidelines. To justify their feeding choices and evade stigma, mothers often resorted to fabricated excuses, and the male partners often supported the mothers’ claims. An effective strategy used by some mothers was defending the feeding choice as the one chosen by the male partner. Such findings were in line with previous research on the influential role that ACB male partners played in infant feeding decisions [46] and the pressures that ACB-mothers LWH often face from relatives, especially in the absence of breastfeeding [6,10,17,25,47]. Therefore, the role of male partners as family protectors is evolving in the context of HIV to encompass protection from stigma, harassment, and discrimination.

In terms of the provision of emotional support, which we believe is a critical safety net to lessen the mental health burdens of HIV on mothers LWH, our study participants noted immense emotional support by their male partners in the form of decision-making help to choose between EBF and EFF, easing the mother’s distress and pain, fending off pressures, as well as overall care and concern for the well-being of the mother and infant. These findings corroborate previous findings in relation to male partners providing emotional support during pregnancy and post-partum, [21,22,23,48,49].

Although the majority of the mothers LWH expressed satisfaction with the support they received from their male partners, a few noted the lack of such support. The lack of male partners’ support complicates the experiences of infant feeding of mothers LWH. Previous studies have shown that a lack of male partner support increases the likelihood of mixed feeding, which can be associated with an increased risk of vertical transmission of HIV [9,13]. Similarly, our findings reflected instances where fathers pressured mothers to mix-feed, as breast milk was believed to be insufficient. This often led to tension, particularly when other family members were involved. In another study, a lack of support from male partners in sub-Saharan Africa was vividly portrayed by the fact that “male partners tend to seek outside sexual partners during this period because women need time to recover from delivery, women focus their attention on the child, and some men are disgusted by breast milk” [40]. This finding was slightly different from the findings in our study, though similarly, some fathers left their women alone to pursue other things like work and drugs. 

Our study also found that a lack of trust among a few of the mothers LWH limited the male partners’ supportive role. Such mothers alluded to a loss of confidence in their partners after the realization that their partner infected them with HIV. In a study conducted by previous studies [50], the findings showed that two participants (mothers) had rejected their partners for infidelity and lack of trust. 

### 4.1. Recommendations

This study strongly recommends the introduction of community awareness programs to educate the communities on the importance of increased social support to mothers affected by HIV and AIDs, and most importantly, emphasizing the critical roles male partners can play to enhance infant feeding and healthy outcomes for both the mother and child. 

This study also recommends increasing couple counseling programs for partners LWH to mentally prepare them of social pressures and challenges which may confront them. This may also entail helping them to accept their statuses and focus on caring for their infant to limit unwarranted antagonism and differences which may complicate the raising of the child. 

### 4.2. Study Limitations

This paper is based on a study that examined the infant feeding practices of Black mothers LWH. The lack of data directly obtained from fathers or male partners is a limitation. As we did not interview the male partners, the views presented were only those of the mothers, and there may have been differences in perspectives between the mothers and the male partners which were not captured in our study. We therefore interpreted our findings bearing in mind this limitation. Moreover, another challenge we faced was finding self-identifying Black mothers LWH in the community; therefore, we resorted to venue-based sampling, where mothers LWH were recruited from healthcare facilities and support groups. 

### 4.3. Future Research

Involving male partners as participants in future studies can help us capture their perspectives on their role in the HIV care and infant feeding practices of their partners, identify specific needs they may have, and ultimately design solutions that can better address these objections for a more successful outcome. This is especially important in rural sub-Saharan settings, where the literature documents the highest resistance on the part of men and grandmothers to modify men’s roles in childcare, and where even project staff are often uncomfortable promoting changes in male roles related to child nutrition [51]. 

## 5. Conclusions

Male partners ease the burden of HIV and infant care among many Black mothers LWH, although a few mothers lack such a supportive role. Although cultural and gender-based stereotypes are strong inhibitors of male partner involvement in infant care, especially in sub-Saharan Africa, Black male partner roles are evolving and expanding to include activities that have been traditionally and culturally ascribed to women. Capturing the perspectives of male partners, as well as launching infant feeding educational programs designed to meaningfully engage women alongside their male partners and various community stakeholders is necessary to enhance HIV care and prevention, especially as it relates to infant feeding practices and the prevention of the vertical transmission of HIV. 

## Data Availability

Data analysis activities are currently underway. To request access to study data, please contact J.E., the principal investigator for the project.

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
