# Peer review of "Male Partners’ Roles in Infant Feeding Practices: Perspectives of Black Mothers Living with HIV in Three Countries"

_healthcare, 2022, doi:10.3390/healthcare10112254_

Round 1

Reviewer 1 Report

OVERALL

Thank you for this interesting paper, which did have some interesting elements, especially around the perceptions of mothers living with HIV of the kinds of helpful support provided by their husbands. 

The paper is well written and structured overall, but I do have some serious reservations about the limited perspective, methodology, and conclusions. 

INTRODUCTION

This section provides a valid background to the topic of challenges ACB mothers living with HIV face when deciding on infant feeding, especially from family or the broader society. 

METHODS

The theoretical background of the study is interesting (intersectionality and postcolonial frameworks) but it wasn’t clear how this affected the data you collected and interpreted. Also, I wasn’t sure how ‘mixed’ the methods were. You state in the limitations section that “community consultation meetings involving community members” were undertaken. How were these done? Who took part?

You state that you undertook 61 semi-structured interviews using a focused ethnographic approach. My view would be that this is stretching ethnography a little – what makes this different to, say, a purely qualitative approach? Also, 61 interviews are a large sample for dealing with purely qualitative data, and whilst I can see the benefit of providing insights in diasporic and indigenous contexts, I do wonder if these contexts are so different that the data can’t be clumped into one single dataset. I also note that some of the data may been collected as long ago as 2016. 

Finally, given that the data are part of a larger study which didn’t explicitly explore the support provided by male partners. You do highlight this in your limitations, but it does detract from the validity of the findings. 

I note that ethical approval was granted. 

RESULTS

Some of these are interesting, and the use of quotations to illustrate your points is helpful. However, I also note that your main argument here is that a supportive partner ensures a suitable and empowering milieu, which you conclude enables the mother freedom to select feeding methods. In my view this is tenuous. 

Also, were there any negative experiences described by the respondents? With such a large sample I would expect that some of the mothers had unsupportive partners refusing to participate or engage helpfully with the mother. How did the mother think this affected her?

DISCUSSION AND CONCLUSION

You relate your findings to what is currently known in the field, and this is helpful, and you include suitable citations and references to other studies. Your limitations section is appropriate, though sadly also includes the key element that would have made the paper much more robust (interviews with male partners). There is also no mention of the inherent limitations in qualitative research – that all data are respondent perceptions only.

REVIEWER RECOMMENDATIONS

1.     Describe in depth the mixed methods approach undertaken for these data – what were the community consultations?

2.     Make more explicit how your theoretical perspective drove the data collection and analysis.

3.     Add more discussion about the impact of supportive partners on the infant feeding choices of women – how strong is this link?

4.   In the results section, include more detail about the negative experiences of mothers which I would expect to be in the data somewhere.

Author Response

Please find attached document with the responses to reviews. 

Reviewer 2 Report

A novel qualitative study conducted.  The strength of the study includes the contribution of male partners involvement to optimal Infant feeding practices of exposed infants and health of ACB women in multicounty setting.

Specific comments 

Citation: well written

Abstract: Well written, concisely summarizes the study sections (Introduction, study objectives, methodology, analysis findings and conclusion.

Recommendation for improvement: Consider including brief recommendation (missing)

Study purpose- Clearly written, briefly stated in the in the abstract and again in more details in the introduction

Introduction section

The statement of research is clear and makes the study important, relevant, and interesting. The justification is clear and applicable in settings where culture has huge impact on infant and young child feeding in the context of HIV.

Major Recommendation for improvement:

·        Introduction , para 2, sentence 3: Exclusive Breastfeeding (WHO 2016) is for 6 months NOT 12 months. Please consider revising accordingly.

·        Demographic characteristics of participants section, para 1, sentence 3, consider moving to Study Limitation section. 

Study design: Appropriate, a qualitative design, research method diverse, includes interviews and FGD using structured guide, and audio technology. Sampling process included combination of purposeful and Snowball. Sampling was continued till redundancy (saturation) in data was reached.

Ethic consideration: Informed consent was obtained and ethical procedure from and multiple ethic boards

Minor Recommendation for improvement: Please include information on how the informed consent was obtained (?verbal/written/recorded)

Data collection: Clear with complete description.

Minor Recommendation for improvement:

1.      Include information that is missing to shed light on how the research can be understood.

2.      Document researchers’ credential and previous experience in interviewing in similar countries to increase confidence of a reader in the process.

Procedual rigour: Procedure used was well described, data gathering process including gain to access to the site and data collection method, organized into codes, categories and/or themes. Findings consistent with and reflective of data

Minor Recommendation for improvement:

·        Include training of the data collectors, length of time spent, amount of data collected Major finding summaries in analysis.

·        The different cultural context, how was question of nuances and reasons behind Knowledge attitude and behavior revealed in the data handled

Audibility:

Minor Recommendation for improvement:

1.      Consider including a mention on use of decision/audit trail to track decision made during the process.

Analysis: method used was mentioned including how data was transformed into codes and themes and interrelationship

Discussion Section: clear description of concept and relationship between concepts and integration of relationships among meanings that emerged from data.

Overall riguor: There is evidence of trustworthiness including use of variety of methods to gather data, and triangulation

Minor Recommendation for improvement: Consider including period of collection of data, use of reflective approaches (journal of reflection), biases preconceptions and ideas

Study limitation:

Minor Recommendation for improvement:

·        Consider including information on strategies used to limit bias in the research (data neutrality, reflective, keeping journal)

·        What general and specific challenges encountered in the multi-country setting

Conclusion: Consistent, congruent with findings and contributes to theory. Helps reader to understand theory developed and provides insight into social network issues of Infant and young child feeding in the context of HIV

Implication and recommendation: Explicit, linked to situation and research direction

Reference: missing in the reference,

Major Recommendation for improvement: Consider including the reference below that was referenced but missed in the reference list:

World Health Organization, United Nations Children’s Fund. Guideline: updates on HIV and infant feeding: the duration of breastfeeding, and support from health services to improve feeding practices among mothers living with HIV. Geneva: World Health Organization; 2016.

Author Response

Please find attached document for response to reviews. 

Round 2

Reviewer 1 Report

Thank you for addressing my comments, which I can see have been incorporated into the paper.

I only have one remaining recommendation - on page 10 final paragraph, final (full) sentence, I would change this phrase:

ORIGINAL: Previous studies have shown that lack of male partners support was consistent with mixed feeding, (a condition highly undesirable) which promotes PMTCTs...

 RECOMMENDED CHANGE: Previous studies have shown that lack of male partners support increases the likelihood of mixed feeding, which can be associated with increased risk of vertical transmission of HIV (Maru et al., 2009; Doherty et al., 2006).

Author Response

The Sentence on page page 10 final paragraph has been changed from "Previous studies have shown that lack of male partners support was consistent with mixed feeding, (a condition highly undesirable) which promotes PMTCTs..." to "Previous studies have shown that lack of male partners support increases the likelihood of mixed feeding, which can be associated with increased risk of vertical transmission of HIV (Maru et al., 2009; Doherty et al., 2006)."

Track changes have been used on the word document to such that changes can be easily viewed.